# Implementation pilot study of community self-testing for COVID-19 among employees of manufacturing industries and their household members in 2022 to 2023

Huan Keat Chan[1], Elena Marbán-Castro[2], Sunita Abdul Rahman[3], Xiaohui Sem[2], Nurul Farhana Zulkifli[2], Suziana Redzuan[4], Alias Abdul Aziz[5], Nurhanani Ayub[2], Paula Del Rey-Puech[2], Elena Ivanova Reipold[2], Olga Denisiuk[2], Norizan Ahmad[1], Othman Warijo[1], Muhammad Radzi Abu Hassan[1], Sonjelle Shilton[2]*

1 Clinical Research Centre, Hospital Sultanah Bahiyah, Alor Setar, Kedah, Malaysia, 2 FIND, Geneva, Switzerland, 3 Kedah State Health Department, Alor Setar, Kedah, Malaysia, 4 Kuala Muda District Health Office, Sungai Petani, Kedah, Malaysia, 5 Kulim District Health Office, Kulim, Kedah, Malaysia

☯ These authors contributed equally to this work.
* sonjelle.shilton@finddx.org

**Data Availability Statement:** The protocol for this study, approved by the Medical Research and

## Abstract

COVID-19 self-testing is essential for enabling individuals to self-care, screen themselves and, if positive, isolate themselves. Since 2021, COVID-19 self-tests have been extensively used in high-income countries, however, their programmatic implementation in low- and middle-income countries has been delayed. An implementation pilot, mixed-methods study, was conducted in four industrial manufacturing companies, in Kedah State between November 2022 and May 2023. Participants were asked to take COVID-19 self-tests home for themselves and their household members and to use the tests according to national guidelines. At enrolment and at the end of the study, participants completed an online sociodemographic, knowledge and satisfaction survey. Data were cleaned and analysed using SPSS Statistics V28.0. Qualitative data were collected through semi-structured interviews and focus group discussions. Thematic analysis was conducted. A total of 1768 employees from four manufacturing industries enrolled in the pilot, representing 60% of the total employees and more than 50% of employees at each site. There were 40 COVID-19-positive cases detected in participants from the manufacturing industries, and 100 positive household members. Participants reported 27 invalid test results. Individuals aged 30 or less [adjusted odds ratio (AOR): 2.65; 95% CI: 1.63 to 4.31; p<0.001] and males (AOR: 1.54; 95% CI: 1.09 to 2.17; p = 0.014) showed a significant higher likelihood of self-testing compared to older and female participants. Additionally, individuals who received three or more doses of a COVID-19 vaccine had higher odds of using self-tests (OR 1.56 (95% CI: 1.03 to 2.36, p = 0.037)). There was a significant increase in participants' knowledge on how to correctly collect a self-sample using a nasal swab from 36,9% at baseline to 43,6% post-implementation (p = 0.004) and correct interpretation of a positive result from 80,5% at baseline to 87,6% post-implementation (p<0.001). Furthermore, there was a notable increase in the correct understanding of actions following a positive result, especially regarding self-isolation,

Ethics Committee, by the Ministry of Health Malaysia, did not include provisions for the public sharing of the full database generated during the study. As participants did not provide consent to sharing their data publicly and to ensure compliance with national ethical guidelines, authors did not include the full database in a public repository. The datasets used and/or analyzed during the current study are available from the corresponding author on reasonable request or at the following email address: info@finddx.org.

**Funding:** This research was funded by the Kreditanstalt für Wiederaufbau ("Credit Institute for Reconstruction") KfW Group, the German state-owned investment and development bank; grant number KfW-TBBU02. The funders had no role in study design, data collection and analysis, decision to publish, or preparation of the manuscript.

**Competing interests:** The authors have declared that no competing interests exist.

which rose from 59.1% to 71.9% (p<0.001). A total of 44 SSIs, and 4 FGDs with a total of 14 participants, were performed. The five main themes explored were: 1) previous experiences with COVID-19, 2) COVID-19 ST experiences during the pilot study, 3) advantages of COVID-19 ST, 4) feelings related to COVID-19 ST, 5) willingness to use COVID-19 ST again, and 6) ST for other diseases. This research shows the feasibility of a self-testing model in the community through workplaces due to participants' high acceptability to enrol and high self-tests' uptake. Lessons learnt can inform operational aspects of the introduction and scale-up of self-care strategies in low- and middle-income countries, in particular the South-East Asia region.

## Introduction

The World Health Organization (WHO) announced that, as of 5 of May 2023, the COVID-19 pandemic no longer met the definition of a Public Health Emergency of International Concern, but that the pandemic itself was not over [1]. Different public health strategies are needed to continue to mitigate the COVID-19 pandemic and to prepare for future health emergencies [2]. COVID-19 self-testing is a complementary diagnostics strategy, essential for enabling individuals to screen themselves and, if they test positive, isolate themselves in a timely manner [3]. As of 8 of July 2023, according to the Malaysian Ministry of Health (MoH), the country registered 5.1 million COVID-19 cases since the beginning of the pandemic, with the highest burden concentrated in young individuals from 18 to 39 years of age [4].

Establishing effective testing, including early diagnosis, isolation and contact-tracing strategies remains a critical component of infectious disease control and the global response to the pandemic [5]. Point-of-care (POC) rapid diagnostic tests (RDTs) are used to facilitate rapid diagnosis for decision-making. Self-testing is a POC strategy that allows individuals to fully perform their tests and interpretation of results. COVID-19 self-tests are widely used [6]. COVID-19 self-tests are being authorised for use in many countries, increasing testing capacity and relieving the burden on healthcare systems. Additionally, when integrated with the healthcare system, self-diagnosis is a tool that can empower individuals to be carers of their own health.

In March 2022, WHO released guidance that strongly recommended the use of self-tests in the form of SARS-CoV-2 antigen rapid diagnostic tests (Ag-RDTs), in addition to professionally administered testing services. Self-testing can be considered a diagnostic and/or a screening tool, depending on the epidemiological situation, appearance of symptoms, recent exposure, and to facilitate linkage-to-care. However, the role and use of COVID-19 self-testing will always need to be adapted to national priorities, guidelines, resources and contexts. The WHO guidance highlights that in certain settings, such as schools and workplaces, self-testing costs should not be borne by individuals. The WHO guidelines further advise on the importance of identifying optimal approaches to deliver COVID-19 self-testing based on epidemiology, identified gaps in testing and the broader response, available resources, and the needs of prioritised population groups. Any service delivery approach or distribution channel (e.g. healthcare facilities, nursing homes, workplaces and educational settings, as well as secondary distribution via peers or household members) must be sufficiently agile to reflect the evolving epidemiology and be adapted to suit the local context and community preferences [3].

During the pandemic, the introduction and scale-up of COVID-19 self-testing in most low- and middle-income countries (LMICs) lagged behind that of high-income countries by several months, contributing to the inequalities in access to COVID-19 testing and vaccines. A

systematic review and meta-analysis found high concordance between self-testing and professional use of Antigen RDTs (kappa 0.91) [7]. Most studies included in the review were conducted in HIC (n = 27) and none in a Low income country [7]. A study in Malawi demonstrated high feasibility and acceptability for self-testing among healthcare workers and the general population, emphasizing the potential scale up self-testing in LMIC and suggested further research to develop optimal delivery strategies [8]. Malaysia was a notable exception in the Global South with regards to access to COVID-19 self-testing. The country's national testing strategy, released by the MoH in November 2021, included COVID-19 self-testing as its basis. As of 19 July 2022, approximately 142 COVID-19 self-tests were registered for use in Malaysia, and the government set a price ceiling per test of 19.90 Malaysian ringgit (approximately USD 4.26) [9]. COVID-19 self-test results could be reported via the MySejahtera mobile application [10]. As the pandemic progressed, Malaysia faced a particular challenge with increased COVID-19 transmission in intergenerational households and in individuals in active employment in workplaces. The implementation of a COVID-19 self-testing distribution model among those in active employment, with the aim of reaching multigenerational families in the community, was thougth to potentially be a feasible and acceptable approach to ultimately guarantee safe spaces for family gatherings, education, work and other reasons.

There literature on COVID-19 implementation models in manufacturing industries is scarce. There is one study conducted in a large-scale construction project in Australia, using a mixed-methods approach, that identified gaps in knowledge, adherence to protocols, logistical issues and sustainability of long-term testing [11]. For the scale up of effective interventions involving COVID-19 self-tests in Malaysia and other countries, it is essential to gather information about the operational aspects of implementing a self-testing model among populations at high risk of exposure, along with a secondary distribution model for their households. Knowledge gaps currently exist around the feasibility and acceptability of implementing self-testing strategies for COVID-19 to increase access to testing among individuals working in environments where they are at high risk of contracting COVID-19. This study forms part of a portfolio of projects, led by FIND, which has included the implementation of COVID-19 self-testing models in Brazil, Georgia, India, and Malaysia. The aim of the study was to assess and continuously improve SARS-CoV-2 self-testing, in terms of distribution, uptake, linkage-to-care, reporting, self-testing demand on counselling services, and other operational considerations of a self-testing model, among actively employed individuals and their household members. The general objective of this pilot study in Malaysia was to assess the feasibility of COVID-19 self-testing and to optimise the delivery model to communities at high risk of COVID-19, by the distribution of self-tests through workplaces. The specific objectives were to: 1) assess the feasibility of a SARS-CoV-2 self-testing model by examining the processes, logistics and capacity of sites to provide self-testing report results; 2) assess self-testing uptake and the reporting of results; 3) assess linkage-to-care; 4) assess participants' knowledge acquisition; and 5) explore participants' acceptability of and satisfaction.

## Methods

### Study design

FIND, together with the Malaysian MoH, Kedah State Health Department, Kuala Muda and Kulim District Health Offices, the Federation of Malaysian Manufacturers (FMM) Kedah/Perlis Branch, and four industrial manufacturing companies, conducted a mixed-methods implementation pilot study to explore the feasibility of a COVID-19 self-testing distribution model, with self-tests delivered via workplaces and secondary distribution to households. The study focused on five areas of the feasibility framework proposed by Bowen et al., including

acceptability, demand, implementation, practicality and adaptation [12]. From April 2022 to June 2022, a needs assessment and gap analysis phase took place to identify examples of best practice; conduct landscape research, stakeholder mapping and engagement; and define user-segments. From July 2022 to October 2022, a formative research phase was conducted to inform the design of the screening model and the creation of the protocol and data collection tools. Implementation took place from November 2022 to May 2023, including training, obtaining participants' written informed consent, distribution of self-tests, and data collection. Enrollment occurred from the 15th of November of 2022 to the 21 of February of 2023.

The self-testing model was flexible, as it was designed to be customisable and adapted to meet the needs of each site. Participants were asked to take nasally administered COVID-19 self-tests home, for themselves and their household members, and to use the tests according to the national guidelines [13, 14]. At enrolment and at the end of the study, participants completed an online sociodemographic, knowledge and satisfaction survey; additionally, during the pilot study, participants completed a questionnaire to report their results. Quantitative data were cleaned and analysed in SPSS Statistics V28.0.

Qualitative data were collected from semi-structured interviews (SSIs) and focus group discussions (FGDs) conducted to explore participants' perceptions around the use of self-testing. Qualitative data were collected as structured notes, in Bahasa Malaysia, in a template document created in Microsoft Word, then translated into English. The translated notes were transferred to Microsoft Excel, and a coding analysis was conducted for in-depth exploration and interpretation of the results, identifying patterns and themes. Triangulation of themes was carried out by comparing notes from the SSIs and FGDs.

## Needs assessment and gap analysis

The needs assessment and gap analysis phase started in October 2021 and lasted to January 2022, with the objectives to understand the key drivers of adoption of a specific diagnostic tool, identify key stakeholders in the diagnostic tool landscape, determine key user-segments and use cases where diagnostics can deliver most impact, engage with the team and define needs and gaps. This phase included research best practice examples and country landscape research, stakeholder mapping and engagement, user segment and use case research, stakeholder alignment (workshop and working groups), that helped us with the study design and during the implementation.

## Formative research

The formative research phase lasted between July 2022 and October 2022, with the objective of refining the study design, especially the data collection tools (surveys). The surveys were piloted in one manufacturing industry. This workplace was chosen because it showed similar characteristics to the sites selected for the implementation. It was located in one of the two selected districts in Kedah participated in the piloting of the surveys. Employees of the selected site for the piloting completed the surveys. Questions and answers were reviewed, and changes were integrated into the survey version before study implementation. Additionally, during the formative research phase, study team members travelled to meet implementation partners. Unstructured conversations took place to inform study design and how best to operationalise and adapt the study to meet the contextual needs. Insights were critical for shaping the study's methodology and the protocol.

## Study implementation

This was a pragmatic, observational pilot study of a COVID-19 self-testing model (at home) for employees of manufacturing industries and their household members in Kedah.

Participants could use self-tests when needed. Participants collected nasal swabs and used self-tests in the form of nasal Ag-RDTs during the months following their enrolment, according to updated local guidelines, which stipulated that individuals are advised to conduct a self-test if they experienced COVID-19-like symptoms, were a contact of a positive COVID-19 case, or in other specified circumstances, such as prior to or after attending a mass gathering. A training session was performed at sites to demonstrate the use of the tests to participants and explain the actions they should take after self-testing. At the end of the session, study staff invited employees to participate in the pilot study (enrolment). COVID-19 self-tests were distributed for participants and their household members (two per person), for use under the circumstances described above. If positive, participants reported the results of their self-test to their workplace. All partners involved in the study collaborated to provide overall coordination at each of the four industrial manufacturing sites.

The COVID-19 self-test device used in this pilot study was the Flowflex[TM] SARS-CoV-2 Antigen Rapid Test (Self-testing), manufactured by ACON Biotech (Hangzhou) Co., Ltd., China. It has been approved by the Medical Device Authority, MoH for use in Malaysia; these devices were considered to expose individuals to minimal risks.

## Settings and participants

Five potential sites for inclusion in the study were recommended by FMM Kedah/Perlis Branch, based on employer interest and being located in one of the two districts (Kuala Muda and Kulim) observed by Kedah State Health Department as COVID-19 hotspots during the pandemic; these are also districts in Kedah where industrial manufacturing companies are concentrated. The Kedah State Health Department then selected four sites for participation, during the formative research phase. Production-based manufacturing industries were defined as those where physical creation and processing of goods occur, involving machinery, production lines, and the transformation of raw materials into finished products. Office-based manufacturing industries were those dedicated to administrative, managerial, and support functions, encompassing activities such as planning, coordination, communication, and financial management.

All employees who wished to participate in the study were required to provide written informed consent, by signing an online informed consent form. Once enrolled participants were given a unique identifier (Study ID number) that was used for all study data collection. No personal identifiers were collected at any stage of the study, ensuring anonymity in data handling and analysis. Inclusion criteria for study participation included being willing to provide informed consent, being aged 18 years or older, and working or volunteering at one of the study sites. No informed consent nor data were collected directly from household members. There were no minors included.

For the SSIs, 36 participants were planned to be interviewed, depending on the saturation point [15–17]. Study staff invited participants to join the SSIs until data saturation was reached. The selection of participants for qualitative interviews was based on purposive convenience criteria. Efforts were made to include participants from various genders, sites, job professions, and other relevant factors. Participants were approached by staff at sites, in person or by phone.

## Data collection and processing

**Quantitative data.**   Data collection depended on sites' engagement and enrolment (Table 1). A structured online survey was administered at enrolment to collect participants' sociodemographic data and their knowledge and satisfaction in relation to COVID-19 self-

**Table 1. Timeline for data collection and implementation by site.**

| Site | Data collection started | Data collection finalised | Total duration of pilot study |
|---|---|---|---|
| 01 | November 2022 | May 2023 | 7 months |
| 02 | January 2023 | May 2023 | 5 months |
| 03 | December 2022 | May 2023 | 6 months |
| 04 | February 2023 | May 2023 | 4 months |

testing. An online reporting survey to disclose COVID-19 self-test results was available to be used throughout the entire implementation period (to report the number of tests used and their results). A structured online survey of participants' reporting, knowledge and satisfaction was conducted at the end of the pilot study. The online surveys were continuously monitored for validity, including review by local stakeholders and responses from participants during the piloting phase. The surveys were originally written in English and then translated into Bahasa Malaysia. Both versions were made available for participants. Modified versions in English can be found in the Supporting Information (S1 Annex and S2 Annex). These versions have been edited for consistency and to facilitate readers' understanding. Participants were not mandated to answer all questions.

**Qualitative data.** Qualitative data were collected by conducting SSIs and FGDs, following a semi-structured guide (S3 Annex and S4 Annex). Participants were selected based on their previous consent to participate in an interview, given at the beginning of the pilot study, where the objectives of this study were explained. Participants were interviewed via telephone. Participants only were included in one SSI or FGD. Between 8 and 15 participants per site were selected for the SSIs, depending on saturation. The SSIs were conducted by one female trained interviewer (N.A., B.Sc.) and lasted between 10 and 20 minutes. The SSI guide included a set of questions that corresponded to 1) sociodemographic context and previous experiences with COVID-19, 2) perceptions of and satisfaction with self-testing, and 3) use of self-tests and the value of the reporting mechanism. Additionally, during the last month of the pilot study implementation, four FGDs were conducted. Site staff members who actively contributed to the implementation of the pilot study at each study site were invited to participate in these FGDs. The FGDs were conducted by two female interviewers (X.S., PhD, and N.F.Z., M.Sc.) and notes were taken by another female interviewer (N.A., B.Sc). The three interviewers were full time committed researchers in this study. All three received training by study staff before data collection. FGDs lasted approximately one hour. Participants were not mandated to answer all questions, and any missing data were not imputed. All interviews and discussion were conducted in Malay language, audio-recorded, and notes were taken. They were transcribed and translated to English. Study staff listened to the recordings to gather more in-depth information, if needed.

## Data analysis

**Quantitative data.** The data were analysed using SPSS V28.0. Categorical variables were presented as frequencies and percentages, while numerical variables were presented as means and standard deviations. Backward stepwise logistic regression analysis was conducted to explore factors associated with the willingness to self-test and to report results. All variables with a p-value less than 0.3 in the univariable analysis were included in the multivariable analysis, and the Hosmer–Lemeshow test was used to assess the goodness of fit for the final model. Changes in perceptions of COVID-19 were also summarized using mean scores, and the differences before and after the project implementation were tested using paired t-tests. The

paired t-test was employed to compare the mean scores before and after pilot implementation. The mean scores were calculated based on Likert scale responses, that were represented as ordinal continuous data for analysis. The McNemar test was employed to examine changes in knowledge about COVID-19 self-testing before and after the project. All statistical tests were considered significant if the p value was <0.05.

**Qualitative data.** For qualitative data analysis, content analysis was used. Qualitative data were collected as notes in Microsoft Word documents and transferred to Microsoft Excel spreadsheets for coding analysis, for the in-depth exploration and interpretation of results, identifying patterns and themes. Thematic analysis, constant comparison among interviews and FGDs and investigator triangulation were applied [18, 19]. Two authors (E.M-C. and N.F. Z.) independently generated codes from the transcripts and fitted into the analytical framework. The coding process involved a combination of inductive and deductive approaches, with pre-existing concepts from the interview guide used to categorise the information (themes were deductive from the guide) and codes derived from the data (inductive codes). Meetings were held with the research team to solve disagreements and to identify and share common points. The guides included a set of questions that corresponded to 1) sociodemographic context and previous experiences with COVID-19, 2) perceptions and satisfaction with self-testing, 3) use of self-tests and value of the reporting mechanism, and 4) exploratory questions about willingness to pay for a COVID-19 self-test. For the publication, quotes were selected to exemplify the themes. The study was conducted and reported in line with the Consolidated Criteria for Reporting Qualitative Research (COREQ) (S5 Annex) [20].

## Potential risks

The study design was considered to present a low risk to participants because the COVID-19 self-test used in the study was already approved for use in Malaysia. The collection of nasal swab samples is also of low risk, therefore the probability of an adverse event occurring to a participant and/or to be associated with the self-tests was very low. The only potential inconvenience associated with the study was the length of time (1 to 2 hours) a participant would need to spend completing study procedures, or a breach in participant confidentiality. However, this risk was minimised by training study staff in ethics and confidentiality issues. Furthermore, participants obtained direct benefits, by being able to self-test at their convenience in their home, reducing their risks and uncertainties regarding COVID-19, while the wider community benefitted from having the opportunity of self-tests for members of participants' households. Participants knew their COVID-19 status within 15 minutes after self-testing and learnt to act accordingly. Counselling was provided, and all individuals who tested positive for COVID-19 were linked to care, according to the national guidelines.

## Ethics approval and consent to participate

The study protocol was approved by the Medical Research & Ethics Committee, MoH Malaysia (Ref. No.: NMRR ID-22-02021-UCR, October 26, 2022). The study was conducted in accordance with the protocol and with the ethical principles derived from the Belmont Report [21], the Declaration of Helsinki [22] and applicable ICH Good Clinical Practice E6 (R2) standards [23]. Written informed consent was obtained from all participants.

## Inclusivity in global research

Additional information regarding the ethical, cultural, and scientific considerations specific to inclusivity in global research is included in the Supporting Information (S6 Annex).

## Results

### Enrolment and sociodemographic data

A total of 1768 participants from four industrial manufacturing companies, representing 60% of their total staff and more than 50% of employees at each site, enrolled in the pilot study (Fig 1). Of these, 1763 participants (99%) completed the enrolment survey, and 674 participants (38%) completed the post-implementation survey (Fig 1).

The highest percentage of participants was observed in the 41 to 50 years age group (32.7%), followed by the ≤30 years age group (27.3%). There were more females (57.7%) than males (Table 2). Just 3.7% of participants were of non-Malaysian origin (Table 2). Most participants had received secondary education (50.5%), and the majority of all participants lived with more than three household members (79.9%) (Table 2). The majority of participants had received three or more doses of COVID-19 vaccine (79.1%), and more than half of participants (56%) had previously been diagnosed with COVID-19 disease (Table 2). Most participants had a family member or close friend who had been diagnosed with COVID-19 (67.6%) prior to the implementation, and 10.1% of these diagnosed individuals had died as a result of COVID-19 (Table 2).

### COVID-19 self-testing perceptions, satisfaction and knowledge

At baseline, most participants (63.3%) were worried about COVID-19, were willing to perform a COVID-19 self-test (77.1%), were willing to report self-test results to their employer or to the national database (79.9%) and understood the benefits of self-testing (81.4%) (Table 3).

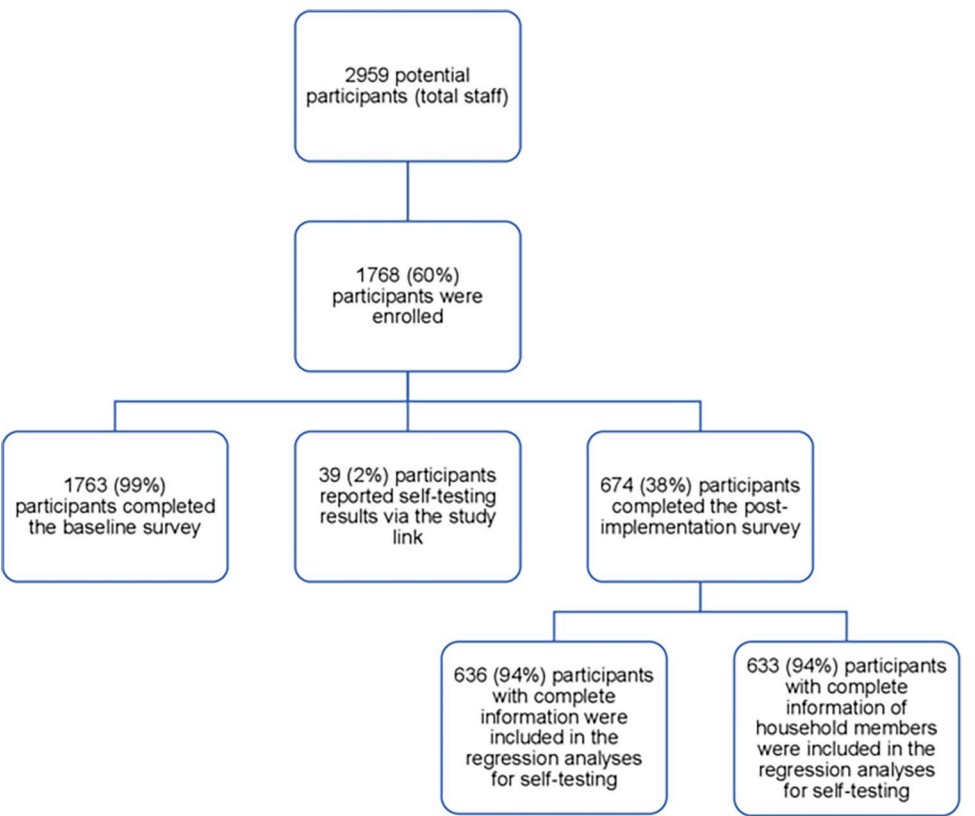

**Fig 1. Flow diagram of study participants in the COVID-19 self-testing implementation pilot in workplaces in Kedah State, Malaysia.**

**Table 2. Sociodemographic characteristics of the participants.**

| Characteristic | n/N (%) |
|---|---|
| Site | |
| 1 (production-based) | 661/1763 (37.5) |
| 2 (office-based) | 254/1763 (14.4) |
| 3 (production-based) | 130/1763 (7.4) |
| 4 (production-based) | 718/1763 (40.7) |
| Age (years) | |
| ≤30 | 482/1763 (27.3) |
| 31–40 | 342/1763 (19.4) |
| 41–50 | 577/1763 (32.7) |
| >50 | 355/1763 (20.1) |
| Undisclosed | 7/1763 (0.4) |
| Gender | |
| Male | 740/1763 (42.0) |
| Female | 1017/1763 (57.7) |
| Undisclosed | 6/1763 (0.3) |
| Country of origin | |
| Malaysia | 1698/1763 (96.3) |
| Indonesia | 20/1763 (1.1) |
| Nepal | 25/1763 (1.4) |
| Myanmar | 10/1763 (0.6) |
| India | 4/1763 (0.2) |
| China | 1/1763 (0.1) |
| Japan | 1/1763 (0.1) |
| Undisclosed | 4/1763 (0.2) |
| Employment status | |
| Full-time worker | 1722/1763 (97.7) |
| Part-time/contracted worker | 26/1763 (1.5) |
| Intern or undergoing industrial training | 12/1763 (0.7) |
| Undisclosed | 3/1763 (0.2) |
| Education level | |
| None | 7/1763 (0.4) |
| Primary | 29/1763 (1.6) |
| Secondary | 891/1763 (50.5) |
| Post-secondary | 517/1763 (29.3) |
| Tertiary | 319/1763 (18.1) |
| Number of household members | |
| 0 | 42/1763 (2.4) |
| 1–2 | 311/1763 (17.6) |
| 3–4 | 764/1763 (43.3) |
| ≥5 | 646/1763 (36.6) |
| Number of household members in employment in the past three months | |
| 0 | 341/1763 (19.3) |
| 1–2 | 1029/1763 (58.4) |
| 3–4 | 321/1763 (18.2) |
| ≥5 | 59/1763 (3.3) |
| Undisclosed | 13/1763 (0.7) |
| Number of household members who are adolescents (aged 12 to 17 years) | |

(*Continued*)

**Table 2.** (Continued)

| Characteristic | n/N (%) |
| --- | --- |
| 0 | 1020/1763 (57.9) |
| 1–2 | 648/1763 (36.8) |
| 3–4 | 64/1763 (3.6) |
| ≥5 | 11/1763 (0.6) |
| Undisclosed | 20/1763 (1.1) |
| Number of household members who are children (aged <12 years) | |
| 0 | 1017/1763 (57.7) |
| 1–2 | 583/1763 (33.1) |
| 3–4 | 124/1763 (7.0) |
| ≥5 | 17/1763 (1.0) |
| Undisclosed | 22/1763 (1.2) |
| Ownership of a smartphone | |
| No | 32/1763 (1.8) |
| Yes | 1720/1763 (97.6) |
| Undisclosed | 11/1763 (0.6) |
| Doses of COVID-19 vaccine received | |
| None | 3/1763 (0.2) |
| One | 8/1763 (0.5) |
| Two | 357/1763 (20.2) |
| Three or more | 1394/1763 (79.1) |
| Undisclosed | 1/1763 (0.1) |
| History and severity of COVID-19 symptoms (in participants) | |
| None/uncertain | 774/1763 (43.9) |
| Yes, unknown severity | 6/1763 (0.3) |
| Yes, asymptomatic (Category 1) | 156/1763 (8.8) |
| Yes, mild to moderate symptoms (Category 2) | 785/1763 (44.5) |
| Yes, severe symptoms (Category 3–5) | 41/1763 (2.3) |
| Undisclosed | 1/1763 (0.1) |
| History and severity of COVID-19 symptoms in the most-affected family member/close friend | |
| None/uncertain | 571/1763 (32.4) |
| Yes, unknown severity | 30/1763 (1.7) |
| Yes, asymptomatic (Category 1) | 155/1763 (8.8) |
| Yes, mild to moderate symptoms (Category 2) | 785/1763 (44.5) |
| Yes, severe symptoms (Category 3–5) | 44/1763 (2.5) |
| Yes, deceased | 178/1763 (10.1) |

Comparing the responses of participants who completed the baseline and post-implementation surveys (N = 671), there were significant changes in perceptions (Table 4). At baseline, 38.5% of participants strongly agreed and 27.4% agreed that they were worried about COVID-19 (Table 4). However, after the pilot study there was a decrease in worry levels, with 29.5% strongly agreeing and 34.1% agreeing that they were worried about COVID-19 (p = 0.028) (Table 4). The only value that was not statistically significant before and after the implementation was the willingness to perform a COVID-19 self-test, with the p-value indicating no significant change (p = 0.243) (Table 4). Before the pilot study implementation, 62.5% of participants strongly agreed and 21.3% agreed to report their self-testing results (Table 4). After the pilot study implementation, these percentages changed to 45.3% and 31.7%, respectively (p < 0.001), indicating a decrease in the willingness to report results (Table 4).

**Table 3. Baseline perceptions about COVID-19 and self-testing and satisfaction around COVID-19 self-testing.**

| Aspect | Baseline data |
|---|---:|
| Worried about COVID-19 | |
| Strongly disagree (1) | 70/1763 (4.0) |
| Disagree (2) | 97/1763 (5.5) |
| Neutral (3) | 469/1763 (26.6) |
| Agree (4) | 431/1763 (24.4) |
| Strongly agree (5) | 686/1763 (38.9) |
| Undisclosed | 10/1763 (0.6) |
| Willing to perform a self-test | |
| Strongly disagree (1) | 66/1763 (3.7) |
| Disagree (2) | 45/1763 (2.6) |
| Neutral (3) | 284/1763 (16.1) |
| Agree (4) | 441/1763 (25.0) |
| Strongly agree (5) | 918/1763 (52.1) |
| Undisclosed | 9/1763 (0.5) |
| Willing to report a self-test result to employer/national database | |
| Strongly disagree (1) | 45/1763 (2.6) |
| Disagree (2) | 42/1763 (2.4) |
| Neutral (3) | 261/1763 (14.8) |
| Agree (4) | 387/1763 (22.0) |
| Strongly agree (5) | 1,021/1763 (57.9) |
| Undisclosed | 7/1763 (0.4) |
| Understand the benefits of self-testing | |
| Strongly disagree (1) | 43/1899 (2.3) |
| Disagree (2) | 36/1899 (1.9) |
| Neutral (3) | 275/1899 (14.5) |
| Agree (4) | 421/1899 (22.2) |
| Strongly agree (5) | 1124/1899 (59.2) |

Participants demonstrated a decrease in their understanding of the benefits of self-testing after the pilot study implementation (p = 0.011) (Table 4). Before the pilot study, 62.0% of participants strongly agreed and 23.7% agreed that they understood the benefits of self-testing; after the pilot study, these percentages changed to 51.2% and 34.7%, respectively (p = 0.011) (Table 4).

Following implementation, there was a significant increase in the percentage of participants who correctly knew how to collect a self-sample using a nasal swab (from 36.9% to 43.6%, p = 0.004) and how to interpret a positive result (from 80.5% to 87.6%, p < 0.001) (Table 5). Conversely, there was a decline in knowledge regarding the interpretation of a faint-line result (from 58.2% to 48.6%, p < 0.001) (Table 5). There were no significant changes in knowledge regarding the correct actions to be taken following a positive test result or the presence of symptoms (Table 5).

## Uptake of COVID-19 self-tests

Of the 674 participants (38%) who completed the survey at the end of the study, more than half of participants (396, 58.8%) and their household members (375, 55.9%), as was reported by participants, used a COVID-19 self-test during the implementation (S7 Annex). In total, 40 participants and 100 household members reported they self-tested positive during the pilot

**Table 4. Changes in perceptions of COVID-19 disease and self-testing before (baseline) and after pilot study implementation.**

| Aspect | Baseline, n/N (%) | Post-implementation, n/N (%) | p-value [a] |
|---|---|---|---|
| Worried about COVID-19 | | | 0.028 (t=-2.20) |
| Strongly disagree (1) | 31/671 (4.6) | 26/671 (3.9) | |
| Disagree (2) | 42/671 (6.3) | 52/671 (7.7) | |
| Neutral (3) | 156/671 (23.2) | 166/671 (24.7) | |
| Agree (4) | 184/671 (27.4) | 229/671 (34.1) | |
| Strongly agree (5) | 258/671 (38.5) | 198/671 (29.5) | |
| Mean (SD) | 3.89 (1.13) | 3.78 (1.07) | |
| Willing to perform a self-test | | | 0.243 (t=-1.17) |
| Strongly disagree (1) | 22/669 (3.3) | 16/669 (2.4) | |
| Disagree (2) | 17/669 (2.5) | 20/669 (3.0) | |
| Neutral (3) | 95/669 (14.2) | 100/669 (14.9) | |
| Agree (4) | 168/669 (25.1) | 210/669 (31.4) | |
| Strongly agree (5) | 367/669 (54.9) | 323/669 (48.3) | |
| Mean (SD) | 4.26 (1.01) | 4.20 (0.96) | |
| Willing to report a self-test result | | | <0.001 (t=-5.93) |
| Strongly disagree (1) | 14/672 (2.1) | 21/672 (3.1) | |
| Disagree (2) | 15/672 (2.2) | 31/672 (4.6) | |
| Neutral (3) | 80/672 (11.9) | 102/672 (15.2) | |
| Agree (4) | 143/672 (21.3) | 213/672 (31.7) | |
| Strongly agree (5) | 420/672 (62.5) | 304/672 (45.3) | |
| Mean (SD) | 4.40 (0.93) | 4.11 (1.03) | |
| Understand the benefits of self-testing | c | | 0.011 (t=-2.56) |
| Strongly disagree (1) | | 14/668 (2.1) | |
| Disagree (2) | | 12/668 (1.8) | |
| Neutral (3) | | 68/668 (10.2) | |
| Agree (4) | | 232/668 (34.7) | |
| Strongly agree (5) | | 342/668 (51.2) | |
| Mean (SD) | | 4.31 (0.88) | |

[a]Paired t-test, performed using data from participants who completed both pre- and post-implementation surveys. SD, standard deviation

study (S7 Annex). Participants reported 27 invalid self-test results. From the simple logistic regression analysis of potential associations between participants' characteristics and the uptake of self-testing, participants aged 30 years or less had a significantly higher likelihood (71.3%) of reporting self-testing use compared with those aged more than 50 years (28.7%), with an odds ratio (OR) of 2.49 (95% confidence interval (CI): 1.54, 4.03; p < 0.001) (S8 Annex). Additionally, individuals in the 31–40 years age group had a higher likelihood (66.7%) of reported self-testing use, with an OR of 2.00 (95% CI: 1.17, 3.44; p = 0.012) compared with those aged more than 50 years (S8 Annex). Moreover, males showed a significantly higher likelihood (64.4%) of reported self-testing use compared with females (35.6%), with an OR of 1.41 (95% CI: 1.00, 1.97; p = 0.047) (S8 Annex). Multiple logistic regression analysis was performed with those variables that had a p-value <0.3 in the simple logistic regression (S8 Annex); these variables were age and gender (Table 6). Younger individuals (in the ≤30 and 31–40 years age groups) had higher odds of reporting self-testing compared with those aged more than 50 years (Table 6). Males were also more likely to engage in self-testing than females (Table 6). Both associations were statistically significant (Table 6). Specifically, individuals aged 30 years or younger exhibited an adjusted odds ratio (AOR) of 2.65 (95% CI: 1.63 to

**Table 5. Changes in knowledge about COVID-19 self-testing before and after implementation.**

| Aspect | Baseline, n/N (%) | Post-implementation, n/N (%) | p-value [a] |
|---|---|---|---|
| Correct method of taking a nasal swab sample (two nostrils, ≤2.5-cm depth) | 248/672 (36.9) | 293/672 (43.6) | **0.004** |
| Correct interpretation of a positive result (likelihood of COVID-19) | 537/667 (80.5) | 584/667 (87.6) | **<0.001** |
| Incorrectly relating a positive self-test result equals a COVID-19 infection in the past | 55/667 (8.2) | 54/667 (8.1) | >0.99 |
| Actions following a positive test result | | | |
| 1. Check updated national guidelines | 397/672 (59.1) | 416/672 (61.9) | 0.263 |
| 2. Self-isolate as much as possible | 412/672 (71.9) | 535/672 (79.6) | **<0.001>** |
| 3. Call close contacts | 385/672 (57.3) | 386/672 (57.4) | 0.99 |
| 4. Wear a mask at work and maintain hygiene measures | 57/672 (8.5) | 50/672 (7.4) | 0.534 |
| Correct understanding of a faint line on a test kit (likelihood of COVID-19) | 388/667 (58.2) | 324/667 (48.6) | **<0.001** |
| Relating a faint line to lower infectivity of COVID-19 | 48/667 (7.2) | 27/667 (4.0) | **0.011** |
| Recommended actions following a negative self-test result in the presence of symptoms (according to national guidelines) | | | |
| 1. Repeat self-testing on the third day of symptoms | 504/673 (74.9) | 554/673 (82.3) | **<0.001** |
| 2. Check the severity of symptoms and consult a doctor if necessary | 411/673 (61.1) | 411/673 (61.1) | >0.99 |

[a]McNemar test, performed on data from participants who completed both pre- and post-implementation surveys.

4.31, p < 0.001) indicating a significant association with the uptake of self-testing. Similarly, males were found to have an AOR of 1.54 (95% CI: 1.09 to 2.17, p = 0.014) in comparison to females. The Hosmer and Lemeshow test confirmed the model's goodness of fit (p = 0.670).

From the simple logistic regression analysis of potential associations between participants' characteristics and the uptake of self-testing among participants and their household members, individuals who received three or more doses of COVID-19 vaccine had higher odds (OR = 1.81, 95% CI: 1.19, 2.73, p = 0.005) of reported self-testing use compared with those who received two or fewer doses (S8 Annex). Individuals who reported having been diagnosed with Category 2 (OR = 1.44, 95% CI: 0.77, 2.67, p = 0.252) COVID-19 severity and those with Category 3–5 (OR = 2.20, 95% CI: 0.67, 7.19, p = 0.192) had higher odds of reported self-testing use compared with individuals with no or uncertain COVID-19 status or severity (S8 Annex). Although the associations were not statistically significant, there appeared to be a trend indicating that individuals with more severe COVID-19 may be more likely to engage in self-testing (S8 Annex). Multiple logistic regression analysis was performed with variables that

**Table 6. Backwards stepwise multiple logistic regression analysis of participants' characteristics associated with their uptake of self-testing.**

| Characteristic | Adjusted odds ratio (95% CI) | p-value [a] |
|---|---|---|
| Age (years) | | |
| ≤30 | 2.65 (1.63, 4.31) | <0.001 |
| 31–40 | 2.13 (1.23, 3.67) | 0.007 |
| 41–50 | 1.30 (0.87, 1.93) | 0.201 |
| >50 | 1 | - |
| Gender, n (%) | | |
| Male | 1.54 (1.09, 2.17) | 0.014 |
| Female | 1 | - |

[a]A Hosmer and Lemeshow test showed p = 0.670.

**Table 7. Backwards stepwise multiple logistic regression analysis of participants' characteristics associated with the uptake of self-testing among their household members.**

| Characteristic | Adjusted odds ratio (95% CI) | p-value [a] |
|---|---|---|
| Country of origin, n (%) | | |
| Malaysia | 1 | - |
| Not Malaysia | 9.05 (1.07, 76.52) | 0.043 |
| Number of doses of COVID-19 vaccine, n (%) | | |
| Two or fewer | 1.85 (1.22, 2.82) | 0.004 |
| Three or more | 1 | - |
| Understand the benefits of self-testing | | |
| Strongly disagree (1) | 1 | - |
| Disagree (2) | 1.42 (0.23, 8.60) | 0.705 |
| Neutral (3) | 3.21 (0.82, 12.52) | 0.093 |
| Agree (4) | 2.35 (0.63, 8.77) | 0.203 |
| Strongly agree (5) | 2.95 (0.81, 10.72) | 0.101 |

[a]A Hosmer and Lemeshow test showed p = 0.880; CI, confidence interval

had a p-value <0.3 in the simple logistic regression analysis of household members (S9 Annex); these variables were country of origin, number of doses of COVID-19 vaccine, and understanding the benefits of COVID-19 self-testing (Table 7). Among individuals of non-Malaysian origin, there was a higher OR of 9.05 (95% CI: 1.07, 76.52) for the uptake of self-testing among their household members compared to Malaysians, indicating a potential association between country of origin and the use of self-tests (p = 0.043) (Table 7). However, it is important to note that the wide 95% CI reflects considerable uncertainty around this estimate, cautioning against a definitive conclusion, given the small number of non-Malaysian participants, which likely contributed to the imprecision of the result. Individuals who received three or more doses of a COVID-19 vaccine also had a significantly higher OR (1.85, 95% CI: 1.22, 2.82) of uptake of self-testing among their household members compared with those who received two or fewer doses of vaccine (p = 0.004) (Table 7). There was no statistically significant association between understanding the benefits of self-testing and the reported use of self-testing (Table 7).

Thirty-nine participants reported their self-test results a total of 57 times via the reporting form. Among them, 53.8% (21/39) were from production-based workplaces, while 46.2% (18/39) were from office-based workplaces (S10 Annex). The most common reason given for the use of a self-test was symptoms (48.7%), followed by attending a mass gathering event (41.0%) (S10 Annex). Two participants (5.1%) reported one positive COVID-19 result. In terms of the number of self-tests used, 66.7% (26/39) of participants used one self-test (S10 Annex).

## Participants perceptions: qualitative data analysis

A total of 44 SSIs, and 4 FGDs with a total of 14 participants, were performed. The sociodemographic characteristics of these participants are provided in the Supporting Information (S11 Annex). The SSI and FGDs yielded into five themes as shown in the Coding Tree (Table 8). The five main themes explored were: 1) previous experiences with COVID-19, 2) COVID-19 ST experiences during the pilot study, 3) advantages of COVID-19 ST, 4) feelings related to COVID-19 ST, 5) willingness to use COVID-19 ST again, and 6) ST for other diseases.

**Theme 1: Previous COVID-19 experiences.** Participants shared that they or their relatives were diagnosed with COVID-19 and used a COVID-19 self-test (either saliva or nasal-

**Table 8. Coding tree of perceptions and experiences explored with participants of the COVID-19 self-testing implementation pilot in Kedah State, Malaysia.**

| |
|---|
| **Sub-theme 1: Previous COVID-19 experiences** |
| 1: Personal |
| 2: Household members |
| 3: Positive |
| 4: Self-testing experiences |
| **Sub-theme 2: Pilot COVID-19 self-testing experiences** |
| 5: Use |
| 6: Behaviour after self-test results |
| 7: Reporting of results for research purposes |
| 8: Reporting of results for national surveillance (through MySejahtera mobile App) |
| 9: Trust in results |
| 10: Changing satisfaction after self-test use |
| 11: Household members' experiences |
| 12: Colleagues' experiences |
| **Sub-theme 3: Advantages** |
| 13: Increased knowledge on ST |
| 14: Increased awareness on COVID-19 |
| 15: Not painful test |
| 16: Easy to use |
| 17: Valuable follow up during implementation |
| **Sub-theme 4: Feelings** |
| 18: Calm |
| 19: Safe |
| 20: Responsibility towards others |
| **Sub-theme 5: Willingness to use again** |
| 21: Will use it again |
| **Sub-theme 6: Self-testing for other diseases** |
| 22: Infections: dengue, tuberculosis, Hepatitis B, influenza, chickenpox, H1N1 |
| 23: Non communicable diseases: diabetes, high blood pressure |

based). Participants compared their experiences with both nasal and saliva tests. One person stated that *"saliva type [self-testing] is not very accurate"* (Female, 25 years old, account officer). Others stated that they prefer the nasal swab because they fear it might affect the results *"you might drink something that might affect your results"* (Female, 33 years old, engineer), or because of the ease of performing the test *"because it is easier to be used"* (Male, 30 years old, engineer).

**Theme 2: Pilot COVID-19 self-testing experiences.** It was noted that participants trusted COVID-19 self-tests, and accepted the use of nasal swabs. For example, one participant declared that *"nasal tests were more accurate, and they were more confident with the result"* (female, 50, senior executive). Participants reported they had used MySejahtera to report their positive self-tests results; however, most stated they felt that there was no need to report negative results. Participants shared how their household members were initially hesitant to use a nasal swab. However, after giving it a try, they discovered that using it was remarkably effortless and even enjoyable. *"At first, [household members] didn't want to use it because it is a nasal swab, but after trying to use it, it's very easy & fun to use"* (Female, 51 years old, officer).

**Theme 3: Advantages of COVID-19 self-testing.** Throughout the pilot study, participants experienced a change in their perceptions regarding COVID-19 self-testing. Initially,

their concerns revolved around the discomfort of having a sample collected for a professional Ag-RDT or polymerase chain reaction test, especially if it involved collection of a nasal swab sample. However, as the pilot study progressed, participants came to realise that self-testing using a nasal swab was very easy to perform. The training sessions also played a crucial role in providing individuals with the necessary guidance and instructions to perform self-testing effectively and correctly. Attending the training, one of the participants stated that "*The training helped to understand how to use the self-test*" (Female, 35 years old, executive). This highlights the positive impact training can have on promoting comprehension and proficiency in the use of self-tests.

**Theme 4: Feelings using COVID-19 self-testing.** Feelings described when asked about their experiences using the COVID-19 self-tests were calmness and safety (knowing that they do not have the infection). This was also related to the responsibility to protect others and the ease of performing the test, as stated: *"Feel safer, feel guaranteed of your own safety. Have to take care of yourself so, as not to infect other people"* (Female, 50 years old, executive).

**Theme 5: Willingness to use again a COVID-19 self-test.** Although weekly testing was not conducted and there were initial concerns about the use of optional self-testing, with individuals relying on their own judgement of their situation or symptoms to decide whether to test, end-of-study results showed continued willingness to self-test, even optionally. One participant expressed the belief that the decision to engage in self-testing varies depending on the situation, suggesting that people would comply and perform the test if it was mandatory: "*If the government would put self-testing mandatory, people would do them [self-testing]. But as testing now is only recommended in case of symptoms, it [self-testing] changes depending on the situation.*" (Male, 39 years old, engineer).

**Theme 6: Self-testing for other diseases.** Participants stated that having self-tests for other diseases would be a good approach at work. They not only mentioned infectious diseases, such as tuberculosis, dengue, flu, hepatitis or chickenpox, but they were also concerned about non-communicable diseases such as diabetes and high blood pressure, as quoted here: *"For diabetes and high blood pressure, good if there is a self-test, can detect early and can always monitor" (Male, 28 years old, executive).*

Both qualitative and quantitative results were presented at the Technical Working Group meeting. There was agreement on the consistency of the data and the findings presented by both methods.

## Discussion

This implementation research was designed to optimise and tailor COVID-19 self-testing approaches to specific contexts and defined user-segments, and to improve service delivery models in workplaces in Malaysia. Our findings provide valuable insights into the factors influencing self-testing perceptions, satisfaction, knowledge and uptake and highlight areas where improvements could be made to enhance the effectiveness and acceptability of self-testing strategies at workplaces. Overall, the use of COVID-19 self-tests was acceptable among participants and household members. Our results are similar to recent studies in the region. In Indonesia, a values and preferences study revealed willingness among respondents to use self-tests and regularly self-test if recommended [24]. Most participants indicated they would self-isolate and communicate a positive self-test result [24]. In Malaysia, a recent nation-wide cross-sectional survey including 1453 responses, showed high willingness (89%) to use COVID-19 self-tests [25]. The common reasons to perform self-testing was to being able to self-isolate (99%), seek treatment (96%) earlier if tested positive. The common reasons not to self-test was to be similar to flu (92%) and having been vaccinated against COVID (78%). In a similar self-testing

program in Canada, 68% of the 116 respondents were very satisfied with the testing program, and 31% were satisfied [26]. Also, almost three quarters (73%) of their participants strongly agree that they would recommend self-testing to others [26].

Regarding sociodemographic characteristics, most participants in our study were employed in production-based sites and were female. Interestingly, younger participants (those in the ≤30 and 31–40 years age groups) reported more COVID-19 self-test use compared with older participants. Gender also played a role in the uptake of self-testing, with males demonstrating a higher likelihood of engaging in self-testing. This gender difference in uptake may have been influenced by a variety of factors, including differences in perception of risk or health-seeking behaviours. Understanding these gender disparities will help in the development of targeted interventions to ensure equitable access to testing opportunities.

Individuals who had received three or more doses of the COVID-19 vaccine had higher odds of self-testing use compared with those who had received fewer doses. This finding suggests that individuals with a higher number of vaccine doses may be more conscious of COVID-19 and perceive self-testing to be an additional measure to ensure their safety. In terms of perceptions and knowledge, while the majority of participants understood the importance of self-testing at baseline, there was a decrease in the perception of worry about COVID-19 after the pilot implementation. This decrease may be attributed to various factors, such as increasing vaccination rates and reduced COVID-19 cases during the study period. However, it is essential to maintain a certain level of awareness among individuals to ensure continued adherence to preventive measures, including self-testing. A decrease in the willingness to report self-testing results was observed after the pilot implementation, which again could be related to the decrease in COVID-19 cases and hence changes in perception regarding the benefits or need to report self-test results. This finding also highlights the importance of addressing barriers to reporting, such as privacy concerns and a low perception of the need to report negative results. A recent study in India, a mixed-methods study to determine the usability of nasal sampling self-tests for COVID-19 in a peer-assisted model among factory workers, suggested that factory workers could accurately conduct the critical steps for a nasal sampling-based test (80.7%) and interpret their results appropriately with high levels of confidence (93.7%) with trained observer-interpreted results [27]. Other studies have reported lower rates of COVID-19 diagnosis in Malaysia, potentially due to a lower average test positive ratio and testing rate [28]. Higher COVID-19 incidences in the Central, South and Sabah regions have been attributed to underlying higher density and mobility of urban populations [29]. Additionally, poorer living conditions have been associated to have further exacerbated transmission in these regions [29].

Changes in participants' knowledge about COVID-19 self-testing before and after implementation were examined. Significant improvements were observed in some aspects. These findings highlight the varying effects of the implementation on participants' knowledge of COVID-19 self-testing, with improvements in some areas and areas of concern in others, and the importance of having targeted communication strategies. There was an increase in the percentage of participants who correctly knew how to use a nasal swab to collect a self-sample and how to interpret a positive result. These findings indicate the effectiveness of the pilot study in enhancing participants' understanding of self-testing procedures. However, there was a decline in knowledge regarding the interpretation of a faint-line result. This result might be attributed to various external factors beyond the scope of the study, including evolving public health messages, the dynamic nature of the epidemic, proliferation of COVID-19 information through press and social media, increased number of self-tests available in the market etc. which may have influenced participants' understanding and perceptions of test results. This result is especially relevant for manufacturers to provide clearer instructions for use regarding the positivity

of the faint line and to other program implementors to provide specific sensitization and continuous education. Further educational efforts should focus on clarifying the meaning of faint-line results and the appropriate actions individuals should take when encountering such results.

The findings from the SSIs and FGDs provided additional insights into participants' perceptions of and experiences with self-testing. Participants expressed trust in self-tests and reported a preference for nasal swabs over saliva-based tests, due to their perceived better accuracy. This finding highlights the importance of offering self-testing options that are perceived to be reliable and accurate, to enhance user confidence. Additionally, participants commonly used the MySejahtera app to report positive self-test results, indicating its utility as a reporting platform. However, there was a perception that reporting negative results was unnecessary, which warrants further investigation to understand the underlying reasons for this perception and to address any misconceptions. Malaysia took the lead in the use of self-testing technology, well in advance of WHO and global recommendations, because of the country's needs, to reduce the healthcare burden during the pandemic and preserve healthcare resources for those in greatest need and to have price ceiling caps; Malaysia also made both saliva- and nasal-based tests available and developed the MySejahtera mobile application for reporting results [10].

Implementing a COVID-19 self-testing programme at workplaces of different sizes and with different levels of resources is feasible. This programme not only provided employees with self-tests but also extended this provision to their household members, relatives and friends in the community. However, communication campaigns must be strengthened to encourage individuals to use self-tests and report their results, even if those results are negative. These findings highlight the importance of effective communications strategies for promoting public health interventions and ensuring accurate reporting of results for national surveillance. Overall, our results suggest that workplace-based self-testing programmes have the potential to improve access to testing in the community and contribute to controlling the spread of COVID-19 and potentially other future pandemics. Self-testing distribution models could be defined according to the type of workplace where they are to be deployed, individuals' previous experiences with self-testing, exposure to infected individuals etc. Strong public and private collaboration, involving the MoH, state health departments, district health offices, and industrial associations, was key for the successful implementation of the pilot study and potentially for pandemic preparedness. Other factors that might influence the ST strategy, and how to adapt and establish different health interventions into workplaces, are based on the type of workplaces (production- vs. office based), time availability of employees, number of employees, previous experiences with COVID-19 and with ST, access to testing, working conditions and digitalization. To introduce and scale-up COVID-19 self-testing in a particular setting or target population, it is important to tailor self-testing packages and screening models to specific populations and optimise self-testing service delivery packages. To achieve this optimisation, it is essential to understand the feasibility and cost of the model and its acceptability among potential end-users. Our study assessed a screening model in a highly controlled way. Beyond the framework of this study, if the model were to be scaled-up, there could be multiple barriers that might prevent individuals from self-testing on a regular basis, reporting positive results and taking responsible hygiene and preventive action as per authorities' recommendations. Such barriers could include individuals being faced with lost days of work, limiting their ability to earn an income, as was noted in previous studies in the region [30]. Findings from qualitative studies conducted in Indonesia and Brazil suggest the use of self-testing could reduce demand on health facilities while addressing many of the usual barriers to the uptake of services, leading to the timely testing of large numbers of individuals [24, 31, 32]. Despite

Malaysia's great success in introducing COVID-19 self-testing during the pandemic, similar to many other countries some gaps remained in ensuring access to testing and care in the community. Furthermore, there was limited evidence accrued on how to effectively design and implement COVID-19 self-testing programmes for specific user-segments, such as workplaces or schools. Reaching all members of the community for COVID-19 testing and screening can be challenging. Lessons learnt from the present study and others have the potential to influence policies on self-sampling, both nationally and internationally. The information resulting from this study will allow the creation of tool kits for the implementation of self-testing, to enable Malaysia and other countries to rapidly deploy and scale-up self-testing programmes. Limitations of this study include potential bias towards acceptability of the implementation (social desirability bias). Social desirability bias refers to the tendency of respondents to answer questions in a manner that will be viewed "favourable" by others, often leading to overreporting "good behaviour" [33]. Additional limitations are the risk of memory bias and observer bias in the interviews performed. Memory bias refers to the distortion of memory over time, that could have potentially influenced how participants recalled some information and report their experiences with self-testing. As most participants did not respond to the reporting survey once they self-tested, but preferred to respond just once at the end of the study for the post-implementation survey, we are aware that recall bias might have occurred, especially in remembering negative results. Observer bias occurs when discrepancies exist between the state of an event and its recording or observation. In our study, it could have influenced the notes taken during the qualitative data collection. Finally, there may some limitations due to self-tests specificity and sensitivity. To assign a diagnosis of COVID-19, users' interpretation of a self-test result must be considered in combination with other information of clinical relevance, such as the appearance of symptoms, exposure to a positive case of COVID-19, and following updated local and national guidelines. Despite some potential limitations, WHO recommends that Ag-RDTs for SARS-CoV-2 should be offered for self-testing, due to the evidence suggesting that users can reliably and accurately self-test and the acceptability and feasibility of self-testing, with the ultimate goal of reducing inequalities in access to testing [3].

Our study provides important insights into the perceptions, satisfaction and knowledge around COVID-19 self-testing, as well as its uptake, among individuals who work for industrial manufacturing companies. The findings suggest the need for targeted interventions to increase self-testing uptake among older individuals, females and those who have received fewer doses of COVID-19 vaccine. Moreover, efforts should be made to address barriers to reporting self-testing results and provide clear guidance on the interpretation of faint-line results. Enhancing public awareness and understanding of the benefits of self-testing and the procedures involved can contribute to the effective implementation of self-testing as a complementary strategy in controlling the spread of COVID-19. Future research should explore the feasibility and acceptability of self-testing in different settings and populations, to further inform evidence-based public health interventions. This includes investigating the potential integration of self-testing kits for multiple diseases and/or digital tools for improved tracking and reporting of infections, mapping disease patterns, and studying the long-term feasibility and effectiveness of self-testing programs in controlling infectious diseases in the community. Knowledge gained from this study may also benefit society more broadly by improving COVID-19 diagnosis in LMICs. Furthermore, the delivery package toolkits used to implement the strategy could be adapted for use in other contexts. The lessons learnt from this study can be used to tailor and optimise self-testing delivery packages and models, drive demand generation for diagnosis and self-testing and support market approval of self-testing devices in jurisdictions where self-tests remain unregulated. The evidence gained will contribute to the

control of COVID-19 as well as build capacity for tackling future pandemics, particularly those that might be caused by respiratory viruses and where self-testing is available.

## Conclusions

This research shows the feasibility of a self-testing model to the community through workplaces due to participants' high acceptability to enrol and self-tests' uptake. The secondary distribution of self-tests to the household members of actively employed individuals led to the detection of more than 300 cases of COVID-19 in the community, potentially reducing the burden on the healthcare system. Therefore, workplace-based self-testing programmes are feasible and have the potential to enhance access to testing in the community and contribute to the mitigation of future pandemics. However, there are still knowledge gaps that need to be addressed to enable individuals to interpret self-test results correctly and understand national guidelines on the correct actions to take following a self-test. This emphasises the importance of optimising self-testing delivery packages for different user-segments and the need to gather evidence about knowledge gaps, so that communications messages and strategies can be tailored for different populations.

## Supporting information

**S1 Annex. Structured survey administered at baseline.**
(DOCX)

**S2 Annex. Structured survey administered at end-point.**
(DOCX)

**S3 Annex. Semi-structured interview guide.**
(DOCX)

**S4 Annex. Focus group discussion guide.**
(DOCX)

**S5 Annex. COREQ (COnsolidated criteria for REporting qualitative research) checklist.**
(DOCX)

**S6 Annex. Checklist: Inclusivity in global research.**
(DOCX)

**S7 Annex. Reported uptake of COVID-19 self-tests.**
(DOCX)

**S8 Annex. Simple logistic regression analysis of potential associations between participants' characteristics and their uptake of COVID-19 self-testing.**
(DOCX)

**S9 Annex. Simple logistic regression analysis of potential associations between participants' characteristics and the uptake of self-testing among their household members.**
(DOCX)

**S10 Annex. Participants' COVID-19 self-test results via reporting survey, overall.**
(DOCX)

**S11 Annex. Sociodemographic characteristics of participants interviewed.**
(DOCX)

## Acknowledgments

We would like to thank the FMM Kedah/Perlis Branch, all study sites and all participants for their contribution to this study. We would also like to thank the Director-General of Health, Malaysia, for his support throughout the conduct of this study and permission to publish the findings. Editorial support was provided by Adam Bodley, according to Good Publication Practice.

## Author Contributions

**Conceptualization:** Paula Del Rey-Puech, Elena Ivanova Reipold, Muhammad Radzi Abu Hassan, Sonjelle Shilton.

**Formal analysis:** Huan Keat Chan, Elena Marbán-Castro, Nurul Farhana Zulkifli.

**Funding acquisition:** Elena Ivanova Reipold, Sonjelle Shilton.

**Methodology:** Huan Keat Chan, Elena Marbán-Castro, Xiaohui Sem, Nurul Farhana Zulkifli, Nurhanani Ayub.

**Supervision:** Huan Keat Chan, Sunita Abdul Rahman, Xiaohui Sem, Paula Del Rey-Puech, Elena Ivanova Reipold, Olga Denisiuk, Muhammad Radzi Abu Hassan, Sonjelle Shilton.

**Writing – original draft:** Elena Marbán-Castro.

**Writing – review & editing:** Huan Keat Chan, Elena Marbán-Castro, Sunita Abdul Rahman, Xiaohui Sem, Nurul Farhana Zulkifli, Suziana Redzuan, Alias Abdul Aziz, Nurhanani Ayub, Paula Del Rey-Puech, Elena Ivanova Reipold, Olga Denisiuk, Norizan Ahmad, Othman Warijo, Muhammad Radzi Abu Hassan, Sonjelle Shilton.

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
