## [Decision Letter · Decision Letter 0]

24 Nov 2023

PGPH-D-23-01835

Implementation pilot study of community self-testing for COVID-19 among employees of manufacturing industries and their household members in 2022 to 2023

Dear Dr. Shilton,

Thank you for submitting your manuscript to PLOS Global Public Health. After careful consideration, we feel that it has merit but does not fully meet PLOS Global Public Health’s publication criteria as it currently stands. Therefore, we invite you to submit a revised version of the manuscript that addresses the points raised during the review process.

The manuscript has been evaluated by two reviewers, and their comments are available below. The reviewers have raised a number of concerns that need attention. They request additional information on methodological aspects of the study (such as the sample size, data collection and statistics), and additional details on the results. Could you please revise the manuscript to carefully address the concerns raised?

We look forward to receiving your revised manuscript.

Kind regards,

Marianne Clemence

Staff Editor

Journal Requirements:

Additional Editor Comments (if provided):

Reviewers' comments:

Reviewer's Responses to Questions

**Comments to the Author**

1. Does this manuscript meet PLOS Global Public Health’s publication criteria? Is the manuscript technically sound, and do the data support the conclusions? The manuscript must describe methodologically and ethically rigorous research with conclusions that are appropriately drawn based on the data presented.

Reviewer #1: Yes

Reviewer #2: Yes

2. Has the statistical analysis been performed appropriately and rigorously?

Reviewer #1: Yes

Reviewer #2: No

3. Have the authors made all data underlying the findings in their manuscript fully available (please refer to the Data Availability Statement at the start of the manuscript PDF file)?

Reviewer #1: Yes

Reviewer #2: No

4. Is the manuscript presented in an intelligible fashion and written in standard English?

Reviewer #1: Yes

Reviewer #2: Yes

5. Review Comments to the Author

Reviewer #1: 1- The message of the article is simple, and the performed study is demonstrative. I consider this manuscript useful for current practice in the field.

2- I think the introduction section can be improved a little bit. It would be better to show the current research progress in this field, so that the research gaps can be more clear. At present, I feel that the presentation of research gaps is a bit subjective.

3- The process and methods are in general well described.

4- Please add more results from other research and compare your results with other authors' results

5- Please provide further information about the study limitation and future research.

6- How did the researchers calculate the sample size?

7- How did the researcher ensure that the survey and the interview guide were valid and reliable?

8- How did the researcher ensure the confidentiality of participants personal information?

9- It would be advisable to elaborate more in the discussion regarding the quantitative phase results with comparison to the published studies in literature.

Reviewer #2: Implementation pilot study of community self-testing for COVID-19 among employees

of manufacturing industries and their household members in 2022 to 2023.

This manuscript addresses a clear and important public health topic. The manuscript is well-written, and the implementation method is clear enough to allow replication in other settings. The findings are also well presented. However, the study has major and minor issues that need to be addressed by the authors before a publication is considered.

Below are concerns that we think need to be addressed the authors:

1. The manuscript’s abstract does not discuss the study results as presented in the result section, it rather focuses on diagnostic yield as if this was a diagnostic or a prevalence study. Findings from the pre-enrolment and the post-implementation interviews and those from the qualitative component should appear in this abstract.

I personally do not see the need of the following paragraph in the abstract: “The study focused on five areas of the feasibility framework proposed by Bowen et al., including acceptability, demand, implementation, practicality and adaptation. The feasibility of an iterative an adaptive COVID-19 self-test delivery model in Malaysia was assessed, to deliver self-tests to the community via workplaces. FIND, together with the Ministry of Health Malaysia, Kedah State Health Department, Kuala Muda and Kulim District Health Offices, the Federation of Malaysian Manufacturers Kedah/Perlis Branch, and four industrial manufacturing companies, conducted a mixed-methods implementation pilot study between November 2022 and May 2023.”

2. The first line of the manuscript reads: COVID-19 self-testing is essential for enabling individuals to perform self-care, screen themselves and, if they test positive, isolate themselves in a timely manner. Is “perform self-care” appropriate here?

3. If I understand well, the industries ‘employers were trained to perform self-tests and not their household members. Given the design of the study, the household members may have self-tested themselves without assistance from the household’s member trained for self-testing. Do we know how often this happened? What was the strategy used to minimise self-tests by untrained study participants?

4. In lines 202 and 203, it is clearly stated that no informed consent nor data were directly collected from households’ members while in Figure 1 last box, it is stated that 633 participants with complete information were included in the regression analysis for self-test in households’ members. In Table 2, the characteristics of these households ‘members are also presented. Did the authors get authorisation to analyse and publish data of study participants without their informed consent?

5. Is it possible to reorganise the flow diagram (Figure1) for it to account for all study participants including households ‘members reached in a comprehensive order? In its current state, I assume that the 39 participants who reported self-test results never completed the baseline and the post-implementation surveys.

6. From line 330 to line 343, the following words are regularly used “some changes, slight decrease, no significant changes” while the p values presented are always way below 0.05.

a. When using the above-mentioned terms, are you alluding to statistical or clinical differences? Because statistically, the differences are significant.

7. The data in table 4 is categorical and not continuous, why was t test used to assess differences?

8. As As per the study flow chart (Fig 1), only 39 study participants reported self-testing results while in lines 361 to 365, it is reported that more than half of the study participants reported using self-tests during the study implementation.

a. Did many people report using self-test but not the test results? If yes, can you provide clear information? How many study participants reported using self-test without reporting the test results and how many reported using self-testing with results?

b. How many people reported negative and invalid self-test results?

c. After how long were the test results supposed to be reported to the study team? Results reported after a long period may have been affected (recall bias) and this should be discussed.

d. What was the process of reporting test results for household members?

9. In results presented from line to 378, interpretation of the odd ratios is more appropriate and would give a clear picture of the analysis. Some of the 95% CI include 1, please be cautious when interpreting the result.

10. What was the rational for using “Hosmer and Lemeshow test”?

11. In the qualitative results, is it possible to avoid terms like “more than half”, instead give the actual numbers e.g., 25/44(57%).

12. From line 505 to line 510, you present factors that may have affected willingness to report COVID-19 results and how this can be addressed, which I think is an important point. Why do you then compare findings from this study to low positive COVID-19 diagnostic reported in another study in Malaysia and to another study reporting high COVID-19 incidences (lines 510 to 514)? How do low or high COVID-19 diagnostics reported in these two studies link or justify a drop in willingness to report COVID-19 self-test results?

13. “There was an increase in the percentage of participants who knew correctly use a nasal swab to collect a self-sample and how to interpret a positive result”. Lines 520 and 521 need review.

14. Throughout the study I did not see where effective communication strategies were tested with a positive impact on self-test. In lines 545 and 546 read: “These findings highlight the importance of effective communications strategies for promoting public health interventions and ensuring accurate reporting of results for national surveillance”. Is this an assumption or finding from this study?

15. There is no discussion on the positive self-test results reported by the study’s participants and their households ‘members despite the statement in lines 522 and 523 that there was a decline in knowledge interpretation of the self-test result. There is a need to discuss the self-test results as they might be affected by many factors.

6. PLOS authors have the option to publish the peer review history of their article (what does this mean?). If published, this will include your full peer review and any attached files.

**Do you want your identity to be public for this peer review?** For information about this choice, including consent withdrawal, please see our Privacy Policy.

Reviewer #1: **Yes: **sabaa saleh alhemyari

Reviewer #2: No

---

## [Decision Letter · Decision Letter 1]

2 Apr 2024

PGPH-D-23-01835R1

Implementation pilot study of community self-testing for COVID-19 among employees of manufacturing industries and their household members in 2022 to 2023

Dear Dr. Shilton,

Thank you for submitting your manuscript to PLOS Global Public Health. After careful consideration, we feel that it has merit but does not fully meet PLOS Global Public Health’s publication criteria as it currently stands. Therefore, we invite you to submit a revised version of the manuscript that addresses the points raised during the review process.

We look forward to receiving your revised manuscript.

Kind regards,

Holly Seale

Academic Editor

Journal Requirements:

Please review your reference list to ensure that it is complete and correct. If you have cited papers that have been retracted, please include the rationale for doing so in the manuscript text, or remove these references and replace them with relevant current references. Any changes to the reference list should be mentioned in the rebuttal letter that accompanies your revised manuscript. If you need to cite a retracted article, indicate the article’s retracted status in the References list and also include a citation and full reference for the retraction notice

Reviewers' comments:

Reviewer's Responses to Questions

**Comments to the Author**

1. If the authors have adequately addressed your comments raised in a previous round of review and you feel that this manuscript is now acceptable for publication, you may indicate that here to bypass the “Comments to the Author” section, enter your conflict of interest statement in the “Confidential to Editor” section, and submit your "Accept" recommendation.

Reviewer #2: All comments have been addressed

Reviewer #3: All comments have been addressed

2. Does this manuscript meet PLOS Global Public Health’s publication criteria? Is the manuscript technically sound, and do the data support the conclusions? The manuscript must describe methodologically and ethically rigorous research with conclusions that are appropriately drawn based on the data presented.

Reviewer #2: Yes

Reviewer #3: Yes

3. Has the statistical analysis been performed appropriately and rigorously?

Reviewer #2: Yes

Reviewer #3: Yes

4. Have the authors made all data underlying the findings in their manuscript fully available (please refer to the Data Availability Statement at the start of the manuscript PDF file)?

Reviewer #2: Yes

Reviewer #3: Yes

5. Is the manuscript presented in an intelligible fashion and written in standard English?

Reviewer #2: Yes

Reviewer #3: Yes

6. Review Comments to the Author

Reviewer #2: Implementation pilot study of community self-testing for COVID-19 among employees

of manufacturing industries and their household members in 2022 to 2023.

The manuscript has significantly improved compared to the first version but still has some minor issues that need to be addressed:

The abstract, though reviewed, still does not talk to the content of the paper:

1. None of the qualitative findings is clearly presented in the abstract. This is a mixed method, right?

2. The quantitative results are said to show significant differences and/or associations, but no statistical tests nor p-values are included in the abstract.

Please see below issues that need to be addressed:

1. Line 38-39: Younger individuals (aged 30 and less), and males, showed a significant higher likelihood of self-testing compared to older and female participants.

Please add the test used to assess the association and the result (p value).

2. Line 40-41: Additionally, individuals who received three or more doses of a COVID-19 vaccine had higher odds of using self-tests.

Please add the odds results and 95% CI.

3. Line 41-42: Significant changes in participants’ perceptions and knowledge about

COVID-19 self-testing was noted.

Did you do any statistical significance test for change in knowledge? If no, please review this sentence and avoid “significant”.

4. Line 42-44: There was an increase in participants’ knowledge on how to correctly collect a self-sample using a nasal swab (from 36,9% to 43,6%) and interpret a positive result (from 80,5% to 87,6%).

Please add the test used to assess difference between 36.9% and 43.6%, and 80.5% and 87.6% and the results (p value). If you did not do statistical test, please avoid the term “significant”.

5. Line 209: “that” may need to be removed.

6. Line 222 to 223, please name the figure.

7. Line 346 to 360, please specify test for each p value.

8. Table 4: I am still not convinced that t test is the appropriate test for the data in this table:

1. In the table, I see Mean (SD). What is the meaning of the mean? How did you calculated the mean using categorical data?

2. There are appropriate tests for agreement using categorical data. The data you have in this table is categorical.

9. Line 365 to 371, for each p value please specify the test used.

10. Line 418-421: Among individuals of non-Malaysian origin, there was a significantly higher OR (9.05, 95% CI: 1.07, 76.52) of uptake of self-testing among their household members compared with Malaysians, indicating a strong association between country of origin and the use of self-tests (p = 0.043) (Table 7).

In this case I would not interpret the result as a strong association due to a very wide 95% CI and a p-value that is nearing 0,05. The wide 95% CI tells us that, the result is imprecise (it can as well be 1.08 instead of 9.05 or 76.4). This is because the number of non-Malaysians is very low in the study, and the result does not allow us to draw any conclusion.

Please be cautious when interpreting this result.

11. Line 516-518 need review: “Participants in a similar self-testing program in Canada showed that 68% if tge 116 individuals that responded to the survey were very satisfied with the testing program and 31% were satisfied”.

Reviewer #3: The authors have addressed the comments of the previous reviewers.

7. PLOS authors have the option to publish the peer review history of their article (what does this mean?). If published, this will include your full peer review and any attached files.

**Do you want your identity to be public for this peer review?** For information about this choice, including consent withdrawal, please see our Privacy Policy.

Reviewer #2: **Yes: **Bulemba Katende

Reviewer #3: **Yes: **Mohammed Owais Qureshi

---

## [Editor Report · Decision Letter 2]

3 May 2024

Implementation pilot study of community self-testing for COVID-19 among employees of manufacturing industries and their household members in 2022 to 2023

PGPH-D-23-01835R2

Dear MPH Shilton,

We are pleased to inform you that your manuscript 'Implementation pilot study of community self-testing for COVID-19 among employees of manufacturing industries and their household members in 2022 to 2023' has been provisionally accepted for publication in PLOS Global Public Health.

Best regards,

Holly Seale

Academic Editor
